# Monoclonal Antibody-Based Immunosensor for the Electrochemical Detection of Chlortoluron Herbicide in Groundwaters

**DOI:** 10.3390/bios11120513

**Published:** 2021-12-13

**Authors:** Anaïs Surribas, Lise Barthelmebs, Thierry Noguer

**Affiliations:** 1Biosensors Analysis Environment Laboratory, Université de Perpignan Via Domitia, F-66860 Perpignan, France; anais.surribas@etudiant.univ-perp.fr; 2Biodiversity and Microbial Biotechnologies Laboratory, USR 3579 Sorbonne Universités (UPMC), F-66650 Banyuls-sur-Mer, France

**Keywords:** chlortoluron, immunosensor, screen-printed electrodes, competitive detection, chronoamperometry

## Abstract

Chlortoluron (3-(3-chloro-p-tolyl)-1,1-dimethyl urea) is an herbicide widely used in substitution to isoproturon to control grass weed in wheat and barley crops. Chlortoluron has been detected in groundwaters for more than 20 years; and dramatic increases in concentrations are observed after intense rain outbreaks. In this context; we developed an immunosensor for the determination of chlortoluron based on competitive binding of specific monoclonal antibodies on chlortoluron and immobilized biotinylated chlortoluron; followed by electrochemical detection on screen-printed carbon electrodes. The optimized immunosensor exhibited a logarithmic response in the range 0.01–10 µg·L^−1^; with a calculated detection limit (LOD) of 22.4 ng·L^−1^; which is below the maximum levels allowed by the legislation (0.1 µg·L^−1^). The immunosensor was used for the determination of chlortoluron in natural groundwaters, showing the absence of matrix effects.

## 1. Introduction

The use of pesticides has exponentially increased since their creation to improve crop yields, in response to a constant increase in the world population [1,2]. Currently, the sale of pesticides remains stable at around 350,000 tons per year in the European Union, of which 23,000 tons are for France (EUROSTAT 2019 [3]). Herbicides account for about 48% of the total pesticides usage [4]. Agricultural practices associated with the use of herbicides lead to the contamination of environmental resources, and more specifically groundwaters. The transfer of herbicides to groundwaters occurs mainly through run-off and leaching of agricultural soil and is affected by many factors relative to the chemical nature of active substances, soil composition, and climatic conditions [5]. Since groundwater is one of the main resources for drinking water production, the knowledge of herbicide transfers and fate in the environment is essential to assess and reduce the potential risks. Various indicators able to describe the environmental impact of pesticides have been developed in Europe, which are useful for regulatory purposes to mitigate the use and sale of herbicides [6]. According to the Herbicide Resistance Action Committee (HRAC), herbicides are classified according to their mode of action into 23 groups and subgroups. Phenylurea herbicides (PUs), commercialized since 1950 and classified into HRAC group C2, are known to induce photosynthesis inhibition [7] by blocking the electron transfer at the level of the D1 protein of the photosystem II. PUs are selective herbicides mostly used for pre- or post-emergence control of annual grasses and broad-leaved weeds in cereal, fruit, and cotton fields [8]. Due to their persistence and moderate mobility, most PUs present a high risk of groundwater contamination [9]. Among them, chlortoluron (3-(3-chloro-p-tolyl)-1,1-dimethyl urea) (Figure 1), developed by Ciba Geigy in 1969 [10] and approved by the European Commission on 1 March 2006, is nowadays widely used in France instead of isoproturon to control grass weed in cereal, cotton, and fruit crops [11].

Chlortoluron has been detected in groundwaters for more than 20 years, and dramatic concentration increases have been observed during years with the most rainfall events [12]. Recently, concentrations of 0.14 µg·L^−1^ were reported in surface waters close to oyster farms in the Pertuis Charentais (France) [13]. Many regulations and directives have been established by the European Commission to monitor and control water quality. The Directive 98/83/EC (EC 1998), which concerns the quality of water intended for human consumption, sets threshold values of 0.1 µg·L^−1^ for individual pesticides and their relevant metabolites and 0.5 µg·L^−1^ for the total amount of pesticides [14,15]. Monitoring studies are thus highly desired to verify whether the herbicide concentrations exceed these regulatory threshold values. Conventional methods for detecting herbicides in water are based on chromatographic techniques including gas chromatography and high-performance liquid chromatography (HPLC) [16]. Due to the thermal instability of PUs, HPLC has been widely used for their detection in water samples [17,18]. Multiple detection methods coupled with HPLC have been reported based on either fluorescence [19] or ultraviolet detection [20]. These techniques offer several advantages, such as sensitivity, limits of detection (LOD) under the nanomolar range, specificity, and possibility of multi-analyte detection. However, they are not suitable for field analysis since they require highly trained personnel and sophisticated equipment [21]. These limitations have led to the emergence of alternative analytical tools in the last decades, including biosensors, which have received the most attention due to their ease of use, low cost, sensitivity, rapidity, and portability, allowing on-site detection [22]. These devices are made of the close association of a sensitive biological element and a physical transducer, allowing the conversion of the recognition event into a measurable signal.

Due to the mode of action of PU herbicides, most biosensors designed for their detection were based on the measurement of photosynthesis inhibition using appropriate biological elements like whole photosynthetic cells, chloroplasts, or thylakoids. For instance, an optical biosensor based on the measurement of the chlorophyll fluorescence of isolated chloroplasts was described for the detection of atrazine and diuron in drinking water at the sub-µg·L^−1^ level [23]. A similar sensor involving *Chlorella vulgaris* microalgae cells was described for the detection of simazine, atrazine, isoproturon, and diuron, but LOD values below the threshold value were observed for only diuron and isoproturon [24]. Three other microalga species (*Dictyosphaerium chlorelloides*, *Scenedesmus intermedius*, and *Scenedesmus* sp.) entrapped in silicate sol-gel matrices were used for simazine, atrazine, propazine, terbuthylazine, and linuron with LOD values 10-fold higher than the legal limit [25]. Site-directed mutagenesis was used to modify the D1 protein QB pocket of *Chlamydomonas reinhardtii* unicellular green alga, allowing the detection of three triazine (atrazine, prometryne, and terbuthylazine) and two PUs (diuron and linuron) at the nM level (sub-µg·L^−1^) [26]. Despite their relative sensitivity, most of the biosensors based on photosynthesis inhibition suffer from poor specificity, due to the fact that both PUs and triazines herbicides are likely to be detected by these devices. To tackle this drawback, immunosensors have been developed coupling exploitation of the remarkable affinity of monoclonal antibodies with the sensitivity of optical or electrochemical detection. For instance, a reusable immunosensor was described for direct detection of isoproturon based on surface plasmon resonance (SPR) detection, with an LOD of 0.1 μg·L^−1^, corresponding to the threshold value [27]. An electrochemical immunosensor was developed for diuron based on competitive detection using Prussian blue-modified gold electrodes coated with a protein-hapten conjugate [28]. A similar system involving screen-printed carbon electrodes (SPCEs) and amperometric detection was also described for detecting isoproturon, with an LOD of 0.84 μg·L^−1^ [29]. An impedimetric immunosensor was later developed based on SPCE modified with gold nanoparticles. This system allowed the direct non-competitive detection of diuron with an LOD of 5.46 μg·L^−1^ [30].

To the best of our knowledge, no immunosensor has been reported to date for the specific detection of chlortoluron, and only a colorimetric competitive ELISA was described showing a cross-reactivity for other PUs (chlorbromuron, isoproturon, and metoxuron) [31]. The aim of this study was to develop an immunosensor based on an indirect competitive format using a new monoclonal antibody specific to chlortoluron. As presented in Figure 2, the method was based on the competitive binding of primary monoclonal antibodies with biotinylated chlortoluron immobilized on the streptavidin-coated surface and free chlortoluron. After elimination of chlortoluron-bound primary antibodies, secondary antibodies labeled with horseradish peroxidase (HRP) were allowed to react with surface-bound primary antibodies, and the binding reaction was revealed in the presence of the HRP co-substrates H_2_O_2_ and 3,3′,5,5′-tetramethylbenzidine (TMB). The formation of oxidized TMB was detected either by colorimetric or electrochemical methods.

An indirect colorimetric immunoassay was first developed, and the method was transferred for electrochemical analysis using SPCE as the transducer. Chronoamperometry was used to investigate the response of the electrochemical immunosensor using the same labeled system. The proposed immunosensor was then used for the detection of chlortoluron in natural groundwater samples.

## 2. Materials and Methods

### 2.1. Reagents and Solutions

A standard solution of chlortoluron at 100 µg·L^−1^ in methanol was purchased from Cluzeau Info Labo (Sainte-Foy-La-Grande, France), and diluted solutions were prepared in PBS 1× buffer. pH 7.4. PBS 1× buffer was obtained by mixing 10 mmol·L^−1^ sodium hydrogen phosphate, 1.76 mmol·L^−1^ monopotassium phosphate, 137 mmol·L^−1^ sodium chloride, and 2.7 mmol·L^−1^ potassium chloride. Electrode treatment solution was composed of sulphuric acid 0.5 mol·L^−1^ and potassium chloride 0.1 mol·L^−1^. The “diazonium” solution was freshly prepared in 0.5 mol·L^−1^ hydrochloric acid solution by mixing 10 mmol·L^−1^ of 4-aminobenzoic acid and 10 mmol·L^−1^ of sodium nitrite. Different concentrations of streptavidin (from *Streptomyces avidinii*) were diluted in HEPES buffer (0.1 mol·L^−1^, pH 8). “TMB liquid substrate” was a ready-to-use solution containing H_2_O_2_ and 3,3′,5,5′-tetramethylbenzidine (TMB) HRP co-substrates. Blocking buffer was composed of either 1% or 3% bovine serum albumin (BSA) in PBS 1×. Microplate washing buffer was prepared with 0.1% Tween 20 in PBS 1×. All the aforementioned compounds were purchased from Sigma Aldrich (St Quentin Fallavier, France). The activation solution was freshly prepared by mixing 0.1 mol·L^−1^ EDC and 25 mmol·L^−1^ NHS in MES buffer (0.1 mol·L^−1^, pH 5.5), purchased from Alfa Aesar.

The synthesis of conjugated chlorotoluron-(PEG2-ethylamine)-biotin was performed in collaboration with Chimiothèque, UMR 5246-ICBMS, Université Claude Bernard Lyon 1 (Villeurbanne, France). Solutions of this compound were prepared in carbonate buffer 0.1 mol·L^−1^, pH 9 containing 0.1 mol·L^−1^ sodium bicarbonate and 0.01 mol·L^−1^ sodium carbonate.

Mouse monoclonal antibodies against chlortoluron (anti-ChlT-mAb) were especially produced for this study by Proteogenix (Schiltigheim, France). The secondary antibody was a peroxidase-bound anti-mouse IgG- antibody produced in rabbit (anti-IgG-HRP Ab) (Sigma Aldrich). Solutions of both antibodies were freshly prepared in PBS 1× containing 1% BSA (Sigma Aldrich).

### 2.2. Groundwater Sample Preparation

Groundwater samples were filtered through 0.2 µm syringe filters (Supor^®^ membrane, hydrophilic polyethersulfone) and stored at 4 °C before use. Each sample was spiked with chlortoluron solution at 100 µg·L^−1^, and diluted successively with filtered water. Diluted samples were stocked at 4 °C in amber glass bottles.

### 2.3. Materials

SPCEs were fabricated using a semiautomatic DEK 248 screen-printing system (Model 248CF; DEK, London, UK). One SPCE was composed of a three-electrode system including a graphite working electrode (4 mm diameter circle), a graphite auxiliary electrode (16 mm × 0.8 mm curved line), and an Ag/AgCl pseudo-reference electrode (5 mm × 1.5 mm straight line). The working and auxiliary electrodes were screen-printed using Electrodag 423 SS graphite paste (Scheemda, Oldambt, The Netherlands) and the reference electrode using Ag/AgCl paste (Acheson Electrodag 6037 SS, Scheemda, The Netherlands).

Electrochemical treatments and measurements were performed using a MULTI AUTOLAB M204 potentiostat/galvanostat (Metrohm, Herisau, Suisse) controlled by NOVA 2.1 Software and a Multi Potentiostat µStat 8000P potentiostat/galvanostat (Dropsens, Oviedo, Spain) controlled by DropView 8400 Software.

Colorimetric measurements were performed with an Epoch 2 Microplate Spectrophotometer (EPOCH2TC, BioTek Instruments, Winooski, VT, USA). Nunc Maxisorp 96-well polystyrene microplates used for colorimetric assays were purchased from Thermofisher Scientific (Waltham, MA, USA).

### 2.4. Development of Chlortoluron Detection Tools

#### 2.4.1. Colorimetric Immunoassay

Firstly, 100 µL of streptavidin solution were deposited in the microplate wells. After 90 min of incubation, the solution was removed, and 100 µL of biotinylated chlortoluron solution were incubated for 60 min at 37 °C. To avoid unspecific interactions, 300 µL of blocking buffer containing 3% BSA were added in each well and incubated for 120 min at room temperature. Then, 150 µL of anti-ChlT-mAb and 150 µL of chlortoluron solutions (assay) or PBS 1× (positive control) were mixed in a microtube and incubated for 30 min at 37 °C. Then, 100 µL of this mixture were added to each well and incubated for 60 min at 37 °C with immobilized chlortoluron. Next, 100 µL of anti-IgG-HRP Ab solution were allowed to react in each well for 30 min. Finally, the binding reaction was revealed using 100 µL of TMB liquid substrate. The absorbance of the oxidized TMB was read at 630 nm after 10 min of incubation at 37 °C. All the incubations were protected from light under orbital stirring at 350 rpm. Washing of microplate wells was performed between each step using 3 × 300 μL of washing buffer. All measurements were done in triplicate.

#### 2.4.2. Electrochemical Immunosensor

Electrode modification was adapted from the protocol described by Istamboulie et al. [32]. Firstly, each electrode was covered with 100 µL of treatment solution and subjected to electrochemical pre-treatment by carrying out 5 consecutive cyclic voltammetry scans between +1.0 and −1.5 V vs. Ag/AgCl at 100 mV·s^−1^. Then, 100 µL of “diazonium” solution were dropped on each electrode and left to react for 5 min at room temperature. Surface 4-carboxyphenyl groups were then generated by linear sweep voltammetry from +0.4 to −0.6 V vs. Ag/AgCl (scan rate 50 mV·s^−1^). In total, 50 µL of the activation solution containing EDC and NHS were then incubated for 60 min, and streptavidin immobilization was obtained after incubation of 20 µL of streptavidin solution for 60 min at 4 °C in the dark. After streptavidin immobilization, 20 µL of biotinylated chlortoluron solution were allowed to react for 60 min at 4 °C on the working electrode surface. In order to saturate unbound reactive groups, 50 µL of blocking buffer containing 1% BSA were incubated for 60 min. Then, 100 µL of anti-ChlT-mAb and 100 µL of chlortoluron solutions (assay) or PBS 1× (positive control) were mixed in a microtube and incubated for 30 min at 37 °C. Then, 50 µL of this mixture were dropped on the working electrode and incubated for 60 min. Next, 100 µL of anti-IgG-HRP Ab solution were allowed to react for 30 min, and 40 µL of TMB liquid substrate were added on the electrode. After 1 min of reaction, a potential of −0.2 V vs. Ag/AgCl was applied for 65 s and the current resulting from the reduction of oxidized TMB was concomitantly recorded. Washing of the electrode surface was performed between each step using 3 × 1 mL of PBS 1×. If not specified, incubations were done at room temperature. All assays were performed in triplicate.

### 2.5. Data Processing for the Determination of the Method’s Sensitivity

As described in the previous section, the signal values resulted either from colorimetric immunoassays or chronoamperometric immunosensors. Signal values were expressed as averages of triplicate assays. Three negative controls were performed by measuring the signal in the absence of streptavidin, immobilized chlortoluron, and primary antibodies, respectively. The positive control was performed in the absence of chlortoluron. For each optimization step, the ratio between positive control and negative control signal values was taken into account to determine the optimum conditions. The signal values of positive controls and assays were corrected from the mean value of negative controls. The percentage of binding (%B/B_0_) was then calculated for each chlortoluron concentration by dividing the corrected signal value of the assay by the corrected value of the positive control.

Calibration curves representing the %B/B_0_ versus chlortoluron concentrations were plotted and fitted using Origin Pro 8.6 software (Origin Lab Corporation, Northampton, MA, USA). The limit of detection (LOD) was defined as the chlortoluron concentration inducing a binding decrease of 20%, with a maximum standard deviation of 7%.

## 3. Results and Discussion

### 3.1. Development of the Colorimetric Immunoassay

#### 3.1.1. Optimization of Reagent Concentrations

Preliminary colorimetric assays were performed to test the affinity of the primary antibody for the biotinylated chlortoluron immobilized in the wells. For this purpose, wide concentration ranges of primary monoclonal antibody and conjugated chlortoluron were tested. Streptavidin and secondary antibody concentrations were set at 5 and 2 µg·L^−1^, respectively (Figure 3).

The measured absorbance values increased when increasing the concentrations of antibody up to 0.3 µg·L^−1^. It was shown that a concentration of biotinylated chlortoluron of 0.02 µg·L^−1^ was sufficient to achieve a convenient coating of microplate wells, as no increase of absorbance was observed using higher concentrations. Maximum absorbance values were observed using concentrations of antibodies higher than 0.3 µg·L^−1^, but significant absorbance values of 2.5 were achieved using a concentration of 0.16 µg·L^−1^, whatever the concentration of biotinylated chlortoluron used. With the aim of developing a competitive assay, the lowest concentrations of biotinylated chlortoluron giving a significant response were then optimized. First, competition assays were thus carried out using antibody at 0.16 µg·L^−1^ and biotinylated chlortoluron at either 0.005 or 0.01 µg·L^−1^. Absorbance values measured after incubation with chlortoluron at concentrations ranging from 0.05 at 50 µg·L^−1^ are shown in Figure 4.

No significant differences were observed using the two concentrations of biotinylated chlortoluron tested. For a chlortoluron concentration of 1 µg·L^−1^, absorbance values of 0.83 and 1.01 (%B/B_0_ of 81.5% and 87.6%) were obtained using biotinylated chlortoluron concentrations of 0.005 and 0.01 µg·L^−1^, respectively, but higher standard deviations were observed using a concentration of 0.01 µg·L^−1^. According to these results, subsequent competition tests were carried out using concentrations of 0.005, 0.16, and 1 µg·L^−1^ for biotinylated chlortoluron, monoclonal antibody, and secondary antibody, respectively.

#### 3.1.2. Immunoassay Detection of Chlortoluron

A range of free chlortoluron concentrations between 0.05 and 500 µg·L^−1^ were tested to determine the sensitivity of the proposed immunoassay. The %B/B_0_ ratio was calculated based on the absorbance values measured for each chlortoluron concentration, as described in Section 2.5. The absorbance values obtained for the positive and the three negative controls (mean value) were 1.19 and 0.07, respectively. A decrease in absorbance was observed while increasing the chlortoluron concentration, starting at 1 µg·L^−1^ (OD = 1.08, %B/B_0_ = 90.3%). The calibration curve (Figure 5) was drawn and fitted by non-linear regression using the following four-parameter logistic equation (Origin Pro 8.6 software):(1)$$y=A_{2}+\frac{A_{1}-A_{2}}{1+{\left(\frac{x}{x_{0}}\right)}^{p}}$$
where *y* is the %B/B_0_ for each chlortoluron concentration, *A*_2_ is the highest percentage of binding value, *A*_1_ is the lowest percentage of binding value, *x* is the chlortoluron concentration, *x*_0_ is the EC_50_ value (chlortoluron concentration inducing a 50% decrease of the signal), and *p* is the slope at the inflexion point of the sigmoidal curve [33].

A LOD of 2.7 µg·L^−1^ and EC50 of 8.8 µg·L^−1^ were calculated as the chlortoluron concentrations leading to %B/B_0_ values of 80% and 50%, respectively, using the following equation:(2)$$\%B/B_{0}=0.32+\frac{99.79-0.32}{1+{\left(\frac{LOD}{8.760}\right)}^{1.18}}$$

Even though the sensitivity of the immunoassay did not allow the critical value of 0.1 µg·L^−1^ to be reached, these first results validated the format of the competitive method proposed in this work.

#### 3.1.3. Cross-Reactivity Study

In order to assess its specificity, the optimized immunoassay was tested in the presence of several herbicides including another substituted phenylurea (diuron), a substituted urea (tebuthiuron), a sulfonylurea (triflusulfuron), three chloroacetamides (metazachlor, dimethachlor, pethoxamid), and a diazine (bentazone). The potential cross-response was tested using pesticide concentrations of 0.5 and 5 µg·L^−1^. Figure 6 shows the relative response obtained for each herbicide compared to chlortoluron. As expected, due to its very similar structure, diuron responded in the same manner as chlortoluron, especially when using a high concentration (5 µg·L^−1^). However, the risk of cross-reaction in real analysis is minimized since diuron was banned for agricultural use since 2003. Concerning the other molecules tested, a significant relative response was observed only when using metazachlor and dimethachlor at 5 µg·L^−1^. However, such a concentration is not commonly found in groundwaters.

### 3.2. Development of the Electrochemical Immunosensor

#### 3.2.1. Electrochemical Characterization of TMB on SPCE

In the immunosensor format, the detection of HRP-labeled secondary antibody was based on the electrochemical reduction of TMB oxidized by enzymatic reaction. Cyclic voltammetry (CV) was used to investigate the electrochemical characteristics of TMB on a SPCE, by scanning the potential between +1.0 and −1.0 V vs. Ag/AgCl at a scan rate of 50 mV·s^−1^ (Appendix A). As previously described in the literature, TMB undergoes a two-electron oxidation-reduction process [34], which is characterized in our system by two oxidation peaks at 303 and 540 mV vs. Ag/AgCl, and two reduction peaks at 198 and −50 mV vs. Ag/AgCl. Based on these observations, an applied potential of −200 mV vs. Ag/AgCl was chosen for subsequent chronoamperometric experiments, with the aim of efficiently reducing the oxidized TMB formed on the electrode surface. Such a detection mode was previously described in the literature for the development of immunosensors and genosensors [35,36,37,38,39].

#### 3.2.2. Optimization of Reagent Concentrations

The transfer from immunoassays to immunosensor technology requires optimization of various parameters. Measurements were first performed for optimizing the concentrations of streptavidin and biotinylated chlortoluron used for working electrode modification. These assays were carried out with concentrations of monoclonal and secondary antibody of 3 and 2 µg·L^−1^, respectively. Results showed that using a streptavidin concentration of 5 µg·L^−1^ was sufficient to obtain a correct electrode coating, in such a manner that the ratio between the positive and negative controls was higher than 5, whatever the concentration of biotinylated chlortoluron used (Appendix A). Based on the fact that low concentrations of biotinylated chlortoluron and antibody are mandatory for developing a sensitive immunosensor, concentrations of streptavidin and biotinylated chlortoluron of 5 µg·L^−1^ were selected for carrying out assays using primary antibody at concentrations of 0.6 and 3 µg·L^−1^. Competition was performed in the presence of chlortoluron at either 1 or 10 µg·L^−1^ (Figure 7).

As shown in Figure 7, a dramatic decrease of the sensor response was observed in the presence of chlortoluron at 10 µg·L^−1^, more particularly using an antibody concentration of 0.6 µg·L^−1^. However, such a significant decrease was not obtained using a chlortoluron concentration of 1 µg·L^−1^, which corresponds to the targeted threshold value. In order to improve the sensitivity of the developed immunosensor, additional experiments were performed using lower concentrations of biotinylated chlortoluron (0.1 and 0.05 µg·L^−1^), the concentration of antibody being set at 0.6 µg·L^−1^ (Figure 8). While high standard deviations were observed using electrodes coated with 0.1 µg·L^−1^ of biotinylated chlortoluron, a lower variability was achieved using a concentration of 0.05 µg·L^−1^. In these conditions, chlortoluron at 1 and 10 µg·L^−1^ led to relative binding values (%B/B_0_) of 63.5% and 23%, respectively. Based on these results, the next experiments were carried out with electrodes coated with 0.05 µg·L^−1^ of biotinylated chlortoluron, while competition was performed with a 0.6 µg·L^−1^ antibody solution.

#### 3.2.3. Immunosensor Calibration

Immunosensor measurements were carried out using chlortoluron concentrations ranging from 0 to 10 µg·L^−1^. The chronoamperograms recorded for assays, positive controls, and negative controls are presented in Figure 9A. It can be seen that an increase in the chlortoluron concentration leads to a decrease of the measured current. In the absence of chlortoluron, the intensity of the reduction current was −1.51 µA while for 10 µg·L^−1^ chlortoluron, it was −0.64 µA. Using a concentration of 0.1 µg·L^−1^, the measured current value was −1.17 µA, corresponding to a %B/B_0_ ratio of 71.6%. The average value of the the three negative controls was −0.33 µA. These data allowed a calibration curve representing the %B/B_0_ values versus chlortoluron concentrations to be drawn (Figure 9B), whose equation was obtained by linear regression (least squares method) using Origin Pro software:(3)$$\%B/B_{0}=50.25-7.83\times ln\left[ChlT\right]$$

The above equation allowed calculation of the EC50 (%B/B_0_ = 50%) and LOD (%B/B_0_ = 80%) values of 1.03 µg·L^−1^ and 22.4 ng·L^−1^, respectively. As expected, a highly sensitive response to chlortoluron was achieved with a detection limit approximately 120-fold lower than the colorimetric assay. These results confirmed the potential of the developed immunosensor for detecting chlortoluron at concentrations below the threshold value.

#### 3.2.4. Detection of Chlortoluron in Groundwater Samples

In order to assess potential matrix effects, the developed immunosensor was tested with natural groundwaters. Groundwater samples were first filtered on 0.2 µm membrane and tested before and after being spiked with known concentrations of chlortoluron. The results were compared with those achieved using the buffer as solvent (Figure 10).

In the presence of increasing concentrations of chlortoluron, the value of %B/B_0_ decreased in a similar manner using either buffer or groundwaters as solvent (Figure 10), showing that no matrix effect could be attributed to the two different samples tested. The %B/B_0_ values obtained for 1 µg·L^−1^ chlortoluron were 71.3% and 58.2% for groundwaters 1 and 2, respectively, compared to 73.1% for the buffer. The result obtained for groundwater 2 spiked with 1 µg·L^−1^ chlortoluron could be explained by the presence of very low concentrations of chlortoluron in the original sample, in the µg·L^−1^ order, which could be responsible for the observed discrepancy.

## 4. Conclusions

This work describes for the first time the development of an amperometric immunosensor for the sensitive detection of chlortoluron in waters intended for human consumption. New antibodies were successfully produced and used in combination with biotinylated chlortoluron for the development of a competitive immunoassay. The optimized immunosensor allowed the detection of chlortoluron at concentrations ranging from 0.01 to 10 µg·L^−1^, with an LOD of 22.4 ng·L^−1^. No matrix effects were observed when analyzing natural groundwaters spiked with fixed concentrations of chlortoluron. Considering that the maximal residue limit in drinking waters is 0.1 ng·L^−1^, and taking into account its good performance in terms of sensitivity, the developed biosensor appears to be a promising tool for in-field determination of chlortoluron herbicide.

## Figures and Tables

**Figure 1 biosensors-11-00513-f001:**
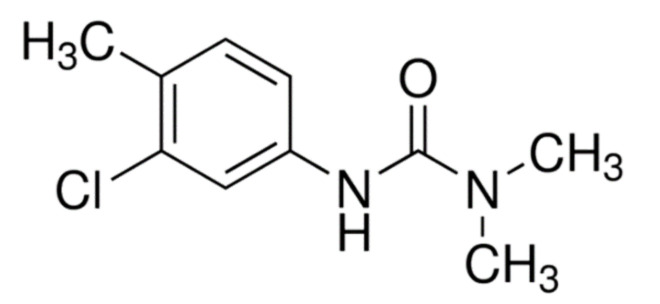
Molecular structure of chlortoluron herbicide.

**Figure 2 biosensors-11-00513-f002:**
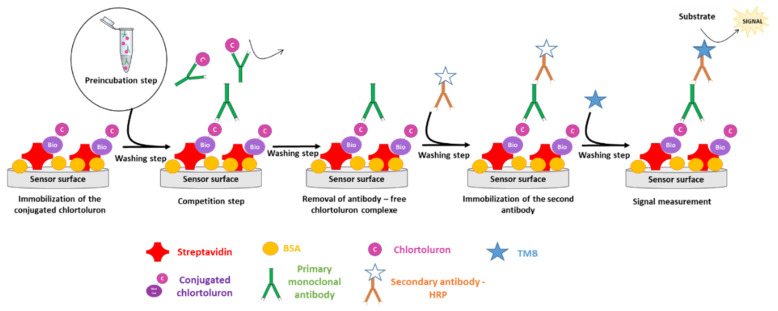
Description of indirect detection based on the competition between immobilized conjugated chlortoluron and free chlortoluron for their binding to primary antibody.

**Figure 3 biosensors-11-00513-f003:**
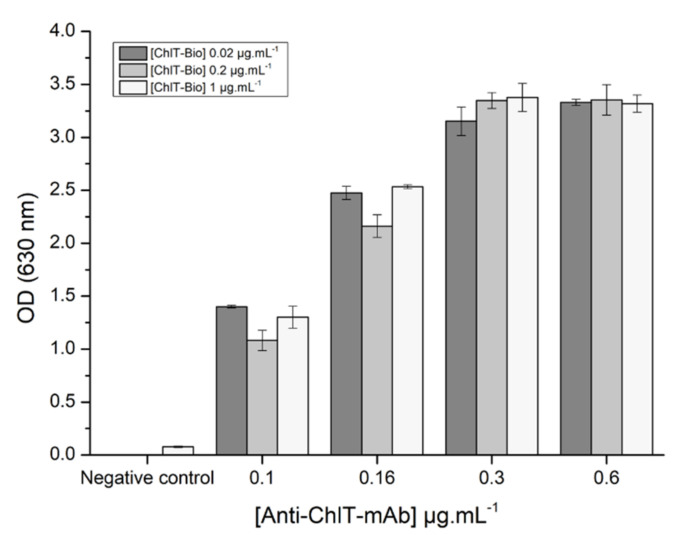
Absorbances measured using microplates coated with biotinylated chlortoluron (ChlT-Bio) at different concentrations (0.02, 0.2, 1 µg·L^−1^) and using increasing concentrations of monoclonal antibody (Anti-ChlT-mAb) (0.1, 0.16, 0.3, 0.6 µg·L^−1^).

**Figure 4 biosensors-11-00513-f004:**
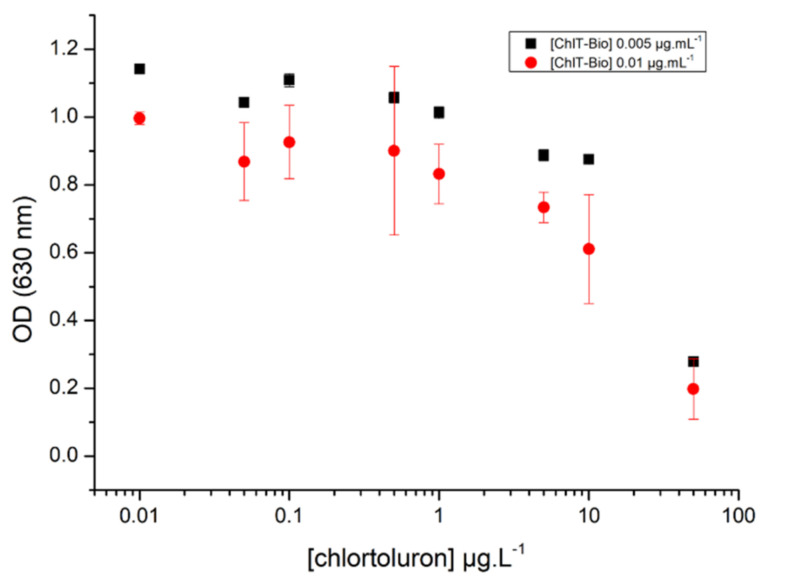
Absorbance measurements at 630 nm obtained after incubation of monoclonal antibody at 0.16 µg·L^−1^ with chlortoluron at concentrations ranging from 0.05 at 50 µg·L^−1^, and using two concentrations of biotinylated chlortoluron (ChlT-Bio) (0.005 and 0.01 µg·L^−1^) (assays in triplicate).

**Figure 5 biosensors-11-00513-f005:**
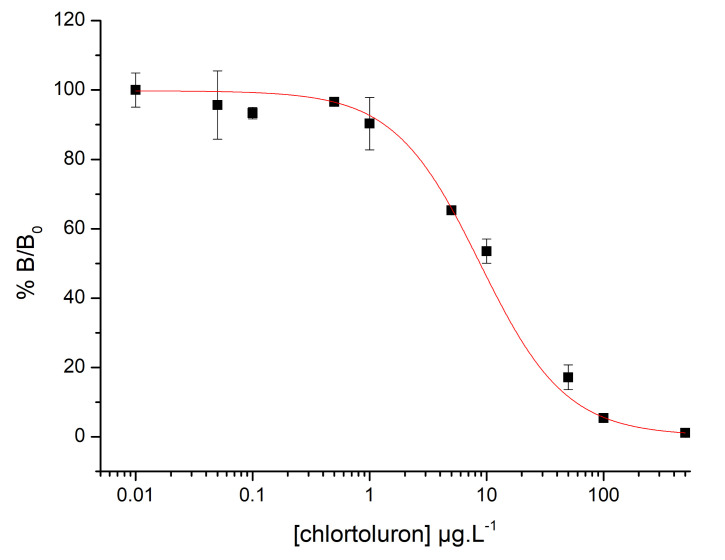
Calibration curve (R^2^ = 0.999) showing %B/B_0_ versus chlortoluron concentrations, obtained with chlortoluron concentrations of 0, 0.05, 0.1, 0.5, 1, 5, 10, 50, 100, and 500 µg·L^−1^.

**Figure 6 biosensors-11-00513-f006:**
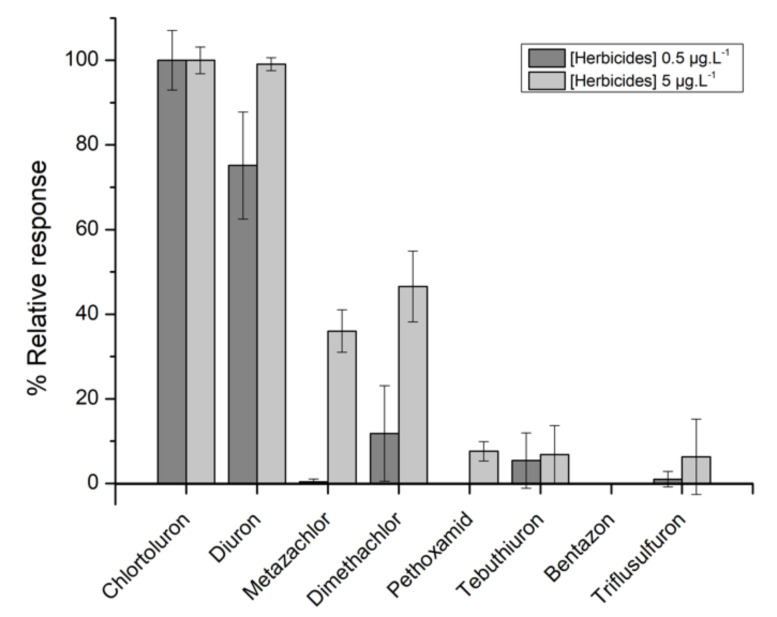
Relative response obtained in the presence of several herbicides compared to chlortoluron at concentrations of 0.5 and 5 µg·L^−1^.

**Figure 7 biosensors-11-00513-f007:**
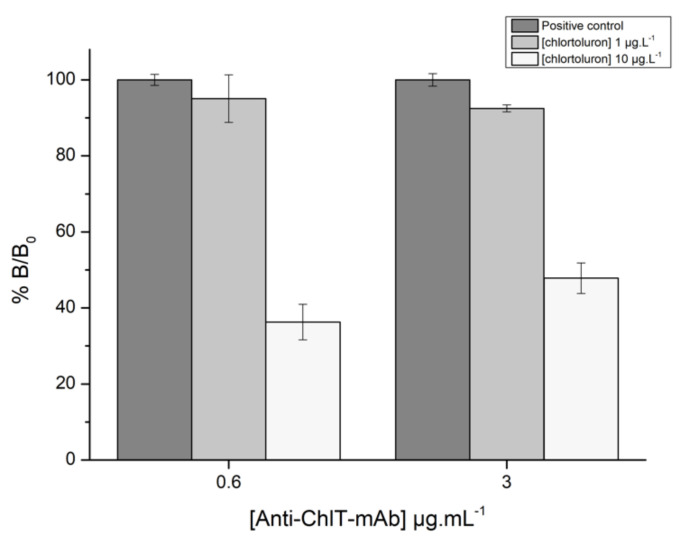
Relative response showing %B/B_0_ obtained in the presence of antibody (Anti-ChlT-mAb) at 0.6 and 3 µg·L^−1^ and in the presence of chlortoluron at 1 or 10 µg·L^−1^. Streptavidin and biotinylated chlortoluron concentrations were set at 5 µg·L^−1^.

**Figure 8 biosensors-11-00513-f008:**
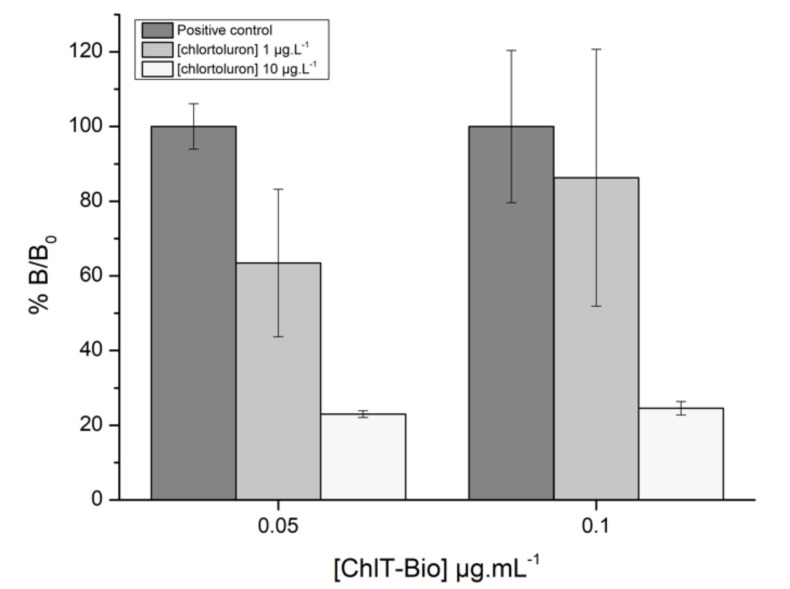
Relative response showing %B/B_0_ obtained for chlortoluron detection (1 and 10 µg·L^−1^) using electrodes coated with biotinylated chlortoluron (ChlT-Bio) at 0.05 or 0.1 µg·L^−1^. Streptavidin and primary antibody concentrations were fixed at 5 and 0.6 µg·L^−1^, respectively.

**Figure 9 biosensors-11-00513-f009:**
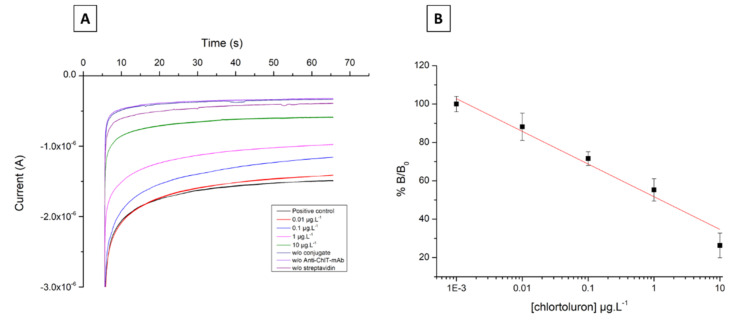
(**A**). Chronoamperograms recorded after incubation with chlortoluron concentrations at 0 (positive control), 0.01, 0.1, 1, and 10 µg·L^−1^. Applied potential −0.2 V vs. Ag/AgCl. Negative controls measured without streptavidin, conjugated chlortoluron, and primary antibody, respectively. (**B**). Calibration curve presenting %B/B_0_ versus chlortoluron concentrations (R^2^ = 0.96).

**Figure 10 biosensors-11-00513-f010:**
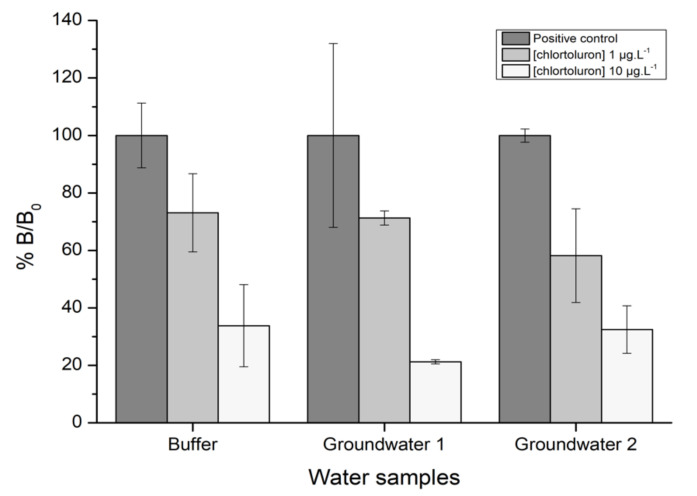
Electrochemical measurements of chlortoluron diluted at 1 and 10 µg·L^−1^ in two groundwaters and in the reference buffer PBS 1× pH 7.4.

## Data Availability

Data available on request due to restrictions.

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
