# Peer review of "Monoclonal Antibody-Based Immunosensor for the Electrochemical Detection of Chlortoluron Herbicide in Groundwaters"

_biosensors, 2021, doi:10.3390/bios11120513_

Round 1
Reviewer 1 Report
The article is well presented, but I would like to suggest a few points:
- Make an abstract graph of the sensor;
- Comparison of electrolyte support solution used with other electrolyte solutions;
- Mention other works that used TMB as an electrochemical mediator (in genosensors and immunosensors);
- On line 204, after 65 is there any data?
-In figure 5, look for 5 points that have linearity and that calculation is based on this equation of the straight line or LOD; In figure 7 look for the smallest standard deviation, to give individuality in the different concentrations;
-In Figure 9 I observe a correlation of the results due to the high standard deviation and thus absence of difference between the results. I suggest redoing the tests the descriptors better the limitations of the study.
Author Response
Reviewer 1
The article is well presented, but I would like to suggest a few points:
Thank you, we have tried to answer your questions properly.
- Make an abstract graph of the sensor;
The sensor principle is explained in figure 2. As explained in the text colorimetric detection was first used to optimize the parameters and then the method was transferred towards electrochemical analysis using SPCE as transducers. We think that introducing an additional figure will not help, but we will follow the editor’s opinion if necessary.
- Comparison of electrolyte support solution used with other electrolyte solutions;
Sorry but we do not really understand the meaning of this question. The electrolyte support used in this work is very classical for immunosensors (PBS 1X).
- Mention other works that used TMB as an electrochemical mediator (in genosensors and immunosensors);
Thank you for this remark. TMB is commonly used in HRP-labelled sensors either as optical or electrochemical mediator. Therefore many other works have used TMB as an electrochemical mediator using screen-printed carbon electrodes and chronoamperometric technique. They include for example immunosensors for the detection of dengue NS1 antigen (Parkarsh 2014), of Streptococcus agalactiae (Vasquez 2017) and imidacloprid pesticide (Pérez-Fernández, 2019). Genosensors were also described for the detection of oligonucleotide sequences (Martín-Fernández 2014) as well as aptasensors (Amaya-Gonzalez 2015). We have included these references in the text (lines 312-314) “Such detection mode was previously described in literature for the development of immunosensors and genosensors [35 – 39]”
- On line 204, after 65 is there any data?
We are sorry for this mistyping, the units (seconds) were missing. We corrected this mistake in the text (65 s).
- In figure 5, look for 5 points that have linearity and that calculation is based on this equation of the straight line or LOD; In figure 7 look for the smallest standard deviation, to give individuality in the different concentrations;
Figure 5 refers to immunoassay detection (with optical measurement). When we take the 5 linear points of figure 5, the obtained equation is y = -18,92ln(x) + 93,38 (R² = 0.9929), the obtained LOD is 2.03 µg/L, which is very close to the one calculated with the whole curve (2.7 µg/L).
The only goal of Figure 7 is to show that using biotinylated chlortoluron at 0.05 µg.mL-1allows enhancing the sensitivity of the immunosensor, while reducing the standard deviation between measurements.
- In Figure 9 I observe a correlation of the results due to the high standard deviation and thus absence of difference between the results. I suggest redoing the tests the descriptors better the limitations of the study.
The observed correlation is not due to the high standard deviation but to the fact that sensor behaviour is the same in buffer and in natural waters. The relative high standard deviations are characteristic of many immunosensors systems, each experiment was realized in triplicate and raw results were considered (no selection of results was made to calculate the mean values). Except for Groundwater 1 positive control, the observed variations were close to the standard values observed in immunoassays and immunosensors.

Reviewer 2 Report
The article biosensors-1478526 entitle “Monoclonal antibody-based immunosensor for the electrochemical detection of chlortoluron herbicide in groundwaters.” describes a new electrochemical competitive immunosensor based on the immobilization of biotinylated chlortoluron (through biotin-streptavidin) interaction. Streptaviding immobilization over carbon screen-printed electrodes is carried out by EDC/NHS using carboxylic groups in the screen-printed working electrode. These carboxylic groups come from the 4-aminobenzoic acid electrografting of the diazonium salt generated in acid medium in the presence of nitrite. The methodology used for electrochemical immunosensor platform development has been widely describes in other works, even in other works published by the same authors. Despite the lack of novelty regarding the immunosensor platform development, it is true that no other electrochemical immunosensor able to detect Chlortoluron has been previously described, so the novelty of the work is only the analyte.
The experiment results are good and authors have optimized different parameters in order to obtain a good concentration range were can applied a logarithmic relation between cathodic current and Chlortoluron concentration. The only experiment that I consider very necessary in order to be published in Bionsensors is an interference study with other similar herbicides, using same family herbicides in order to probe the selectivity of the immunosensor.
Another point to clarify is the result obtained for groundwater 1. As authors are spiking the samples, I suppose the level of Chlortoluron before spiking should be very low, above the LOD. I do not understand why the electrochemical measurement (%B/B0) of groundwater 1 spiked with 10 µg/L is such small comparing with the measured of 10 µg/L spiked buffer and spiked groundwater 2.
Author Response
Reviewer 2
Comments and Suggestions for Authors
The article biosensors-1478526 entitled “Monoclonal antibody-based immunosensor for the electrochemical detection of chlortoluron herbicide in groundwaters.” describes a new electrochemical competitive immunosensor based on the immobilization of biotinylated chlortoluron (through biotin-streptavidin) interaction. Streptaviding immobilization over carbon screen-printed electrodes is carried out by EDC/NHS using carboxylic groups in the screen-printed working electrode. These carboxylic groups come from the 4-aminobenzoic acid electrografting of the diazonium salt generated in acid medium in the presence of nitrite. The methodology used for electrochemical immunosensor platform development has been widely describes in other works, even in other works published by the same authors. Despite the lack of novelty regarding the immunosensor platform development, it is true that no other electrochemical immunosensor able to detect Chlortoluron has been triflusolfuron) previously described, so the novelty of the work is only the analyte.
- The experiment results are good and authors have optimized different parameters in order to obtain a good concentration range were can applied a logarithmic relation between cathodic current and Chlortoluron concentration. The only experiment that I consider very necessary in order to be published in Bionsensors is an interference study with other similar herbicides, using same family herbicides in order to probe the selectivity of the immunosensor.
Thank you very much for this comment. We have taken the time to carry out additional experiments for evaluating the selectivity of the sensor. We tested the cross-response of the device using high concentrations of several pesticides including another phenylureas (diuron), a substituted urea (tebuthiuron), a sulfonylurea (triflusulfuron), 3 chloroacetamides (metazachlor, dimethachlor, petoxamide) and a diazine (bentazone). As expected, due to its very similar structure diuron responded in the same manner as chlortoluron. This cross-reactivity is not a problem since this molecule is banned since 2003.
We have added a sub-paragraph 3.1.3 in the paper to present these results (lines284-296) as well as a figure (new figure 6).
- Another point to clarify is the result obtained for groundwater 1. As authors are spiking the samples, I suppose the level of Chlortoluron before spiking should be very low, above the LOD. I do not understand why the electrochemical measurement (%B/B0) of groundwater 1 spiked with 10 µg/L is such small comparing with the measured of 10 µg/L spiked buffer and spiked groundwater 2.
Thank you for this comment. It is true that the inhibition observed with groundwater 1 spiked with 10 µg/L looks higher that the one observed with buffer or even groundwater 2, even though it must be stressed the standard deviation observed with this measurement was very small compared to the others (for instance the response with buffer was 35±10 and with GW 2 it was 22 ±2). These discrepancies could be explained by the difficulty to properly control the amount of pesticide derivative immobilized on the electrodes, they are commonly found in immunoassays, but we are currently working to ameliorate the immobilization procedure.
Round 2
Reviewer 2 Report
Despite the lack of novelty of the electrodic platform, authors have properly answer my questions and I consider the paper is suitable for publication